



# A Phase Separation Inlet for Droplets, Ice Residuals, and Interstitial Aerosol Particles

Libby Koolik[1,2,3], Michael Roesch[1,4], Lesly J. Franco Deloya[1], Chuanyang Shen[1,5], A. Gannet Hallar[6,7], Ian B. McCubbin[7], Daniel J. Cziczo[1,2,8]

[1]Department of Earth, Atmospheric and Planetary Sciences, Massachusetts Institute of Technology, Cambridge, MA 02139, USA
[2]Department of Civil Environmental Engineering, Massachusetts Institute of Technology, Cambridge, MA 02139, USA
[3]Ramboll, San Francisco, CA 94111 USA
[4]Department of Environmental Systems Science, Swiss Federal Institute of Technology, ETH Zurich, Zurich, 8092, Switzerland
[5]Department of Atmospheric and Oceanic Sciences, Peking University, 100871, Beijing, China
[6]Department of Atmospheric Sciences, University of Utah, Salt Lake City, UT 84112
[7]Storm Peak Laboratory, Desert Research Institute, Steamboat Springs, CO 80488
[8]Department of Earth, Atmospheric, and Planetary Sciences, Purdue University, West Lafayette, IN 47907, USA

*Correspondence to*: Daniel Cziczo (djcziczo@purdue.edu)

**Abstract.** A new inlet for studying the aerosol particles and hydrometeor residuals that compose mixed-phase clouds – the phaSe sePatation Inlet for Droplets icE residuals and inteRstitial aerosol particles (SPIDER) – is described here. SPIDER combines a Large-Pumped Counterflow Virtual Impactor (L-PCVI), a flow tube evaporation chamber, and a Pumped Counterflow Virtual Impactor (PCVI) to separate droplets, ice crystals, and interstitial aerosol particles for simultaneous sampling. Laboratory verification tests of each individual component and the composite SPIDER system were conducted. SPIDER was then deployed at Storm Peak Laboratory (SPL), a mountain-top research facility at 3210m a.s.l. in the Rocky Mountains, for a three-week field campaign. SPIDER performance as a field instrument is presented with data that demonstrates its capability of separating distinct cloud elements and interstitial aerosol particle. Possible design improvements of SPIDER are also suggested.

## 1 Introduction

A mixed-phase cloud has both liquid and ice phases (Korolev et al., 2003; Shupe et al., 2006) with variable number density and mass ratio of liquid to ice particles. Mixed-phase clouds are important factors in aviation and climate (Shupe et al., 2008). In aviation, supercooled droplets can cause aircraft icing and engine power loss (Strapp et al., 2016). In climate, the



role clouds play in the earth's radiative budget remains uncertain (Boucher et al., 2013). As aerosol particle concentration increases in the atmosphere, liquid clouds may have decreased droplet size and increased spatial and temporal extent (Boucher et al., 2013). This will change the radiative forcing at the top of the atmosphere (cloud albedo effect) as well as the lifetime of a cloud (lifetime effect) (Lohmann and Hoose, 2009; Storelvmo et al., 2008). Mixed-phase clouds are particularly complicated because the partitioning of phases is critical in assessing these effects (Atkinson et al., 2013; Hirst et al., 2001;

Korolev et al., 2003; Shupe et al., 2006). At present, these effects are difficult to parameterize in models due to a lack of observational data on formation, properties, and phase partitioning (Kamphus et al., 2010; Shupe et al., 2006). This has resulted in a global effort to study these clouds (Abel et al., 2014; Davis et al., 2007a; Hiranuma et al., 2016; Kupiszewski et al., 2015; Mertes et al., 2007; Patade et al., 2016).

The microphysical formation processes of water and ice clouds are generally understood. Droplets form when a critical

saturation, described theoretically by the Köhler equation, is exceeded. At this saturation aqueous droplets are the favored state and particles that activate are termed cloud condensation nuclei (CCN) (Lohmann and Hoose, 2009; Wang et al., 2012). Ice nucleation is more complex. Ice can form homogeneously, via spontaneous nucleation of ice in a solution droplet, at temperatures below -40°C (Atkinson et al., 2013; Kamphus et al., 2010; Korolev et al., 2003; Storelvmo et al., 2008; Verheggen et al., 2007; Wang et al., 2012). At higher temperatures, ice forms heterogeneously through different pathways

promoted by ice nucleating particles (INPs) (Atkinson et al., 2013; Kamphus et al., 2010; Lohmann and Hoose, 2009; Storelvmo et al., 2008; Tsushima et al., 2006; Verheggen et al., 2007; Wang et al., 2012). The specific properties that determine an effective INP remain poorly understood (Shupe et al., 2008).

There is also uncertainty regarding the existence of both liquid and solid water in the same environment. The accepted theory is the Wegener-Bergeron-Findeisen (WBF) process, whereby ice crystals, depending on the specific environmental

temperature and humidity, grow at the expense of droplet evaporation due to thermodynamic instability (Korolev, 2007; Pruppacher and Klett, 1997). Ice crystals have a lower saturation vapor pressure than water droplets below 0°C, so the presence of crystals will lower the relative humidity and cause the droplets to shrink or, given sufficient time, evaporate completely (Shupe et al., 2006; Storelvmo et al., 2008; Tsushima et al., 2006; Verheggen et al., 2007). This effect is often limited by the concentration of ice crystals in the cloud, since ice is often less than droplet number in mixed-phase clouds

(Verheggen et al., 2007).

In-situ observations are required to understand the natural efficiency of INP and the microphysical processes of mixed-phase clouds. Motivated by climate change, estimated to be warming approximately twice as fast as the global average (Verlinde et al., 2007), several experiments have occurred in the Arctic where there is a prevalence of mixed-phase stratiform clouds (e.g. 41% of the time in the study of Shupe et al. (2006). Another common research location has been the Jungfraujoch, a

mountain-top site in Switzerland, which has high cloud coverage (37% of the time) which are often mixed in phase (Kamphus et al., 2010; Verheggen et al., 2007).

Two of the fundamental questions surrounding mixed-phase cloud formation are: (1) what is the ratio of ice to water in a cloud and (2) what are the aerosol particles that act as the CCN or INPs? Currently, there are a variety of instruments that


can estimate ice or water content of a cloud (Abel et al., 2014; Davis et al., 2007a; Davis et al., 2007b; Korolev et al., 1998;
Strapp et al., 2016), however, these instruments do not report information about the underlying INPs or CCN.

One technique capable of capturing ice and droplet residuals is the Counterflow Virtual Impactor (CVI) and its laboratory
counterpart, the Pumped-Counterflow Virtual Impactor (PCVI). These methods use the idea that activated droplets or ice
crystals are significantly larger than unactivated aerosol particle (Slowik et al., 2011). By separating based on mass,
researchers can study differences between activated and interstitial aerosol particle. This technique has been used in a large
number of studies since the mid-1980s when it was first described by Ogren et al. (1985).

The PCVI uses vacuum-pumped air to form a stagnation plane based on the design of the CVI (Boulter et al., 2006;
Hiranuma et al., 2016). A schematic of the PCVI used in this study is shown in Figure 1. A vacuum pump is used to provide
the "pump flow" (PF), while pressurized air is introduced as an "add flow" (AF). AF has also been referred to as the
"counterflow"; these terms are synonymous with AF used throughout this work.  At the entrance of the PCVI is the "input
flow" (IF) and at the terminus is the "sample flow" (SF) (Boulter et al., 2006; Friedman et al., 2013a). The "effective
counterflow" (ECF) is the difference of AF and SF and counteracts the IF to create a stagnation plane that particles of
sufficient inertia must cross to be entrained in the SF. The 50% cut size or "D50" describes the smallest particle with
sufficient inertia to be transmitted through the PCVI with 50% efficiency. The AF to IF ratio can be adjusted to change the
D50, reducing or increasing the inertial barrier (Kulkarni et al., 2011; Slowik et al., 2011).

Since the original characterizations by Boulter et al. (2006) and Kulkarni et al. (2011), the PCVI has been used in several
cloud sampling studies (Baustian et al., 2012; Friedman et al., 2013a; Slowik et al., 2011). A recent advance is the ability to
build a PCVI using three-dimensional (3D) stereolithography (SLA) printing (Koolik, 2017). 3D printing allows rapid
prototyping for complex devices (Jacobs, 1992), making the development of less expensive PCVIs with higher tolerance
dimensions possible. 3D printing mitigates costs, decreases build time, reduces misalignment, and allows for rapid and
inexpensive tests of potential structural improvements (Koolik, 2017).

## 2 Instrument Theory and Design

SPIDER is a vertically-aligned inlet system with three distinct outlet channels for sampling interstitial (or 'unactivated')
aerosol particles, droplet residuals, and ice crystal residuals (Figure 2). It is comprised of three main components: L-PCVI,
droplet evaporation chamber, and PCVI. The droplet evaporation chamber is actively cooled and lined with a series of
sensors to provide real-time information on the temperature profile.

A 3D printed L-PCVI was based on the design of the machined IS-PCVI described by Hiranuma et al. (2016). When
operated with a 70 L min$^{-1}$ IF and 7 L min$^{-1}$ AF (AF-to-IF ratio of 0.1), the IS-PCVI has a D50 of ~9 μm (Hiranuma et al.,
2016). By operating the L-PCVI with a 50 L min$^{-1}$ IF and 11.5 L min$^{-1}$ AF (AF-to-IF flow ratio of 0.23), the D50 is 20-30
μm. Because droplets and ice crystals are typically 10 μm or larger (Kleinman et al., 2012; Pruppacher and Klett, 1997;



Rogers and Yau, 1989), only these activated droplets and ice crystals are large enough to exit the SF. It should be noted that any droplets and/or ice crystals below the cut size will be stopped and transmitted into the PF.

For the majority of tests described here, the L-PCVI flow rates used in SPIDER were PF, AF, and SF at 55.0, 11.5, and 6.5 L min$^{-1}$, respectively; the PCVI PF, AF, and SF at 8.0, 2.5, and 1.0 L min$^{-1}$, respectively. To ensure that large droplets and ice crystals were transmit without breakup, the Weber Number, NWe, was calculated using these flows. Details and model

results are provided in the Supplement.

Ice crystals and supercooled droplets that pass through the L-PCVI enter the droplet evaporation chamber, which utilizes the WBF process. The chamber can be ice coated and held at -16° C, where the difference in saturation vapor pressure between water and ice is at its maximum. This allows ice crystals to maintain their initial size whereas droplets evaporate during passage. A PCVI is mounted below the droplet evaporation chamber. For this work a commercial machined PCVI (Model

8100, BMI Inc.), described by Boulter et al. (2006) and Kulkarni et al. (2011), and a 3D printed PCVI, described by Koolik (2017), were both used and their performance compared. In SPIDER, the PCVI was operated with D50 ~5.2 μm in order to reject evaporated droplet residuals into the PF but admit ice crystals into the SF. The aforementioned NWe calculations suggest that droplets and ice crystals are not subject to breakup in either the L-PCVI or PCVI.

Using this methodology, SPIDER offers simultaneous sampling channels for interstitial aerosol particle, droplet residuals,

and ice crystal residuals via the PF of the L-PCVI, PF of the PCVI, and the SF of the PCVI, respectively.

## 3 Methodology

### 3.1 3D Component Fabrication

SPIDER incorporates a number of parts that were 3D printed. SLA printing involves the photopolymerization of a liquid resin by a laser in a layer-by-layer process. This printing method was chosen for resolution, surface quality, low shrinkage,

and low distortion (Bartolo, 2011; Bhushan and Caspers, 2017; Hagiwara, 2004). There are drawbacks and common errors that occur with SLA, including overcuring (solidified material fails to bind with the layer below it) and time-intensive post-processing (Jacobs, 1992; Wong and Hernandez, 2012) and parts with these errors were rejected before use. The printer used for SPIDER components (Form 2, Formlabs Inc.) uses a 405 nm UV laser to cure specific coordinates in a resin bath to create the part in a layered structure (3D Printing with Desktop Stereolithography). Parts for SPIDER were printed from

'tough resin' (FLTOTL03, Formlabs Inc.) with a 100-μm layer resolution.

The following procedure was utilized for creating 3D printed parts: A part was virtually designed using Solidworks. The Solidworks assembly tool was used to verify that multiple parts would fit together as designed. Parts were then exported as a stereolithography file (.STL) and uploaded to the 3D printing software (Preform, Formlabs Inc.) where the part was oriented and printing support structures were added. The contact points attached to the part were small (0.5 mm) to ease removal after

printing. From Preform, the part was uploaded onto the printer. After prints were completed, the parts were post-processed following the procedure described by Roesch et al. (2017). Briefly, they were shaken in a bath of isopropyl alcohol (IPA)





and then left for approximately 20 minutes to remove uncured resin. Second, parts are exposed to UV light inside a curing box containing the same wavelength as the printer's laser. Third, the support structures were removed and the surface wet-sanded to the final finish.

### 135 3.2 Instrumentation

Optical particle counters (OPC-N2, Alphasense) with sizing ranges from 0.4 to 17 µm were installed on the PF of the L-PCVI and PCVI (i.e., the interstitial aerosol and droplet residual channels). A higher resolution optical particle sizer (OPS 3330, TSI Inc.) with sizing range from 0.3 to 10 µm was installed on the PCVI SF (i.e., the ice residual channel). Both instrument types feature a monochromatic light source and a size resolution over 16 bins. The OPC has a total flow rate of 140 1.2 L min$^{-1}$ whereas the OPS draws 1 L min$^{-1}$.

The evaporation chamber was cooled using a low-temperature cooling bath (Proline RP 1290, Lauda-Koenigshofen). Additional specifications and the operating procedure for SPIDER are included in the Supplement.

### 4 Verification Experiments

In order to validate the SPIDER method, individual components were tested in the laboratory to determine performance 145 and/or for comparison to previous studies. Droplets or ice crystals were then sent through the complete SPIDER setup to determine transmission, evaporation/sublimation, and rejection efficiency of each phase. It was not possible to form a mixed-phase cloud within the laboratory; a field experiment, detailed in the next section, was used as a final validation step.

### 4.1 L-PCVI

Hiranuma et al. (2016) described the expected working conditions of the L-PCVI at different flow ratios. The operating ratio 150 of AF to IF in L min$^{-1}$ under laboratory conditions in SPIDER is approximately 0.23, so the D50 of the L-PCVI is expected to be 20-30 µm. The OPC and OPS detect particles up to 17 and 10 µm, respectively, so a characterization of the L-PCVI in the style of the Boulter et al. (2006) characterization of the PCVI was not possible. Instead, measurements over a range of expected D50 was used to show that the L-PCVI rejects particles as described in Hiranuma et al. (2016).

To find the lower bound of the D50, 10 µm polystyrene latex (PSL) spheres were atomized (Roesch and Cziczo, 2020). 155 Particles were first introduced to the L-PCVI with PF and AF off to provide the initial particle size distribution. The flows on the L-PCVI were then turned on. The L-PCVI flow values are given in Table 1. Samples were taken at 1 Hz resolution in both the PF and SF of the L-PCVI.

Figure 3a and 3b show the number concentration obtained in the PF and SF, respectively. The location of the measurement is represented by the circle on the inset schematic. The L-PCVI flows are initially off (represented by "X"), and particles are 160 only measured in the SF. The AF is turned on (represented by the directional arrow) approximately 1,900 seconds into the measurement and the PF 100 seconds later. When the L-PCVI flows are off, a detectable concentration of 10 µm particles is



present in the SF; when the L-PCVI flows are on, these particles appear in the PF (i.e., they are unable to cross the L-PCVI stagnation plane and are rejected). We conclude that the cut size of the L-PCVI is larger than 10 μm with the difference in concentration due to the dilution from adding filtered air as AF.

The second experiment replicated performance measurements by Hiranuma et al. (2016) using an IF of 50 and an AF of 7 L min$^{-1}$ (AF/IF ratio 0.14) with a sample flow of 2 L min$^{-1}$. Hiranuma et al. (2016) report a D50 of 22 μm for these conditions. To create particles larger than this expected D50, droplets were generated using a commercial droplet generator (DG) (MD-K-130, Microdrop Technologies) with a 50 μm nozzle. The droplet size leaving the nozzle was verified to be 50 μm diameter using a high-resolution camera and image analysis software. For laboratory conditions and a transit distance of 5 mm from

DG to L-PCVI droplets partially evaporate and are calculated to be 40 μm on entrance to the IF; a model of droplet evaporation was considered based on the equations in Lohmann et al (2016). Details of the model and the results are included in the Supplement. Droplets were created from an ammonium sulfate (AS) solution of 5 g L$^{-1}$. The residuals size of 50 μm droplets of this concentration after complete evaporation is calculated to be 3.7 μm.

Measurement of droplet transmission and residual size were then performed. The stream of droplets was introduced through

the L-PCVI without PF or AF on (Figure 4a). The OPS was connected to the L-PCVI SF with a flow splitter, such that 1 L min$^{-1}$ entered the OPS and 1 L min$^{-1}$ bypass flow to a mass flow controller (MFC) to total the sample flow of 2 L min$^{-1}$. Particles were detected in the OPS 4.7 μm size bin and extend to the maximum size (10 μm), indicative of transmission and partial evaporation. The L-PCVI flows were turned on (Figure 4b). Particles from 4.7 – 10 μm continued to be observed, suggesting the droplets were transmitted through the L-PCVI and partially evaporating. We conclude that the particles

detected in the SF (Figure 4b) are the result of droplets large enough to pass through the stagnation plane. These experiments bracket the D50 of the L-PCVI between 10 and 40 μm, consistent with 22 μm for Hiranuma et al. (2016). In the Supplement, the results from this work are superimposed on the data from Hiranuma et al. (2016).

## 4.2 Droplet Evaporation Chamber

### 4.2.1 Droplet Evaporation

Droplet evaporation was considered based on the aforementioned equations in Lohmann et al (2016). From the model (see Supplement), it is expected that droplets entering the chamber 12.5 μm in diameter or smaller will fully evaporate before reaching the PCVI for chamber saturation 0.9 and lower. Droplets between 12.5 - 25 μm diameter will evaporate if the chamber saturation is below 0.5. Droplets larger than 25 μm are expected to partially but not fully evaporate within the chamber; this sets an effective upper limit for SPIDER.

In practice, the relative humidity (RH) of the chamber and the L-PCVI AF determine the saturation droplets experience. A static SPIDER at -16° C with ice coated walls has a relative humidity with respect to ice that is, by definition saturation, but ~85% RH with respect to liquid water (i.e., the model suggests droplets somewhat larger than 12.5 μm in diameter will fully



evaporate). Hygrometer measurements show that dry air from the L-PCVI AF reduces this to ~75% RH. At this lower RH, the model suggests that droplets 20 μm diameter and smaller fully evaporated in the chamber.

### 4.2.2 Sustaining Ice Crystals

Ice crystals were passed through the chamber to validate transmission. The L-PCVI was removed and 50 μm diameter droplets generated from a 0.6 g L$^{-1}$ AS solution were introduced directly at the top of the chamber. The chiller was set to -65° C, below the homogeneous freezing threshold of ~-38° C (Koop et al., 2000). This initiated rapid homogeneous freezing, validated visually. Ice crystals passed down the chamber to the PCVI which was run with flows to achieve a D50 of 2.7 - 3.8 μm: PF, AF, and SF set to 8.0, 2.5, and 1 LPM, respectively (Figure 5). The resulting particle concentration in the SF of the PCVI, the ice crystal residual flow, is equivalent to the initial concentration of (frozen) droplets. At the initial solution concentration this corresponds to a residual size of AS of 1.4 μm diameter. The residual peak, ~3.8 μm diameter (Figure 5), suggests ice crystals pass through the stagnation plane of the PCVI but do not sublimate/evaporate fully at the OPS.

### 4.3 PCVI

A performance validation of a 3D Printed PCVI was performed by Koolik (2017) following Boulter et al. (2006) and Kulkarni et al. (2011). Using a bubble burst generator ("bubbler") containing a solution of 0.1 g/L ammonium sulfate, measurements of D50 under various flow scenarios were performed and compared. With a constant AF of 2.5 LPM and SF of 1.0 LPM, the 3D printed PCVI had a working range of IF from 3.9 to 9.2 LPM. The results of the comparison within this range are shown in Figure 6.

Using the aforementioned SPIDER flows, a PCVI D50 of ~5 μm diameter is expected from the literature (Boulter et al., 2006; Kulkarni et al., 2011). To validate the D50, the SF from the PCVI was compared to the initial size distribution (i.e., for each size bin of OPS data). The transmission efficiency of each bin size was calculated and the data fit with a sigmoid; the D50 was defined as the particle diameter size that corresponded to 50% of the maximum transmission efficiency on the sigmoid. An example of data using the flows in the last paragraph and the sigmoidal fit corresponding to a D50 of 5.1 μm is shown in Figure 7. The operational flows used SPIDER are summarized in Section 2. Additional verification experiments are summarized in the Supplement.

## 5 Storm Peak Laboratory Field Campaign

### 5.1 Storm Peak Laboratory Field Site

The Desert Research Institute's Storm Peak Laboratory (SPL) is located near Steamboat Springs in north-western Colorado at 40.45°N, 106.74°W, 3210 m above sea level (Borys and Wetzel, 1997). During the winter, SPL is often enshrouded in clouds that contain supercooled liquid and ice crystals, making it an ideal site for sampling cloud condensation nuclei (CCN) and ice nucleating particles (INP) (Friedman et al., 2013b; DeMott et al., 2010). SPL contains a measurement suite for



aerosol particles, cloud properties, and meteorological instruments that provide additional information about ambient conditions. This made SPL an ideal site for deploying SPIDER to test its efficacy for sampling mixed-phase clouds.

## 5.2 Field Measurements

SPIDER was deployed at SPL for three weeks during January 2019. This deployment allowed for performance data of the inlet within mixed-phase clouds to complement the individual component, droplet and ice crystal testing conducted in the laboratory and detailed in the preceding sections. Data from January 20th, 21st and 23rd are presented here to show performance in clear, transition and cloudy conditions. SPIDER was connected to an ambient inlet to access aerosol and cloud elements. The inlet has a cut size of 13μm for at wind speeds of 0.5 m s$^{-1}$ (Petersen et al., 2019) and the L-PCVI of SPIDER was set to a cut size of 8μm; these cut points were set to allow transmission of interstitial aerosol particles, cloud droplets, and ice crystals but reduce large wind-blown ice fragments, discussed in detail in the following paragraphs.

As previously described, two OPCs were attached to SPIDER to measure interstitial aerosol particles and cloud droplet residuals. An OPC on the PF from the L-PCVI is termed the interstitial aerosol channel. An OPC on the PF of the PCVI is termed the cloud droplet residual channel. Both of these OPCs suffered intermittent failures during the field study. The scattering signal from a Droplet Measurement Technologies (DMT) Single Particle Soot Photometer Extended Range (SP2-XR) was used to measure ice crystal residuals from the SF of the PCVI (termed the ice crystal residual channel). The SP2-XR allowed for higher sampling frequency and sizing resolution (100 nm - 540 nm) than the OPCs used in the laboratory but with a smaller size limit. Analysis concentrates on SP2-XR data which was available for the entirety of the study.

Temperature during the study ranged from -5 to -20 °C and wind speed ranged from 0-10 m/s (see Supplement). Background aerosol particle conditions were typically low, with occasional spikes in particle concentrations attributed to new particle formation and local sources (see Supplement). Aerosol spikes were minimal during the three-day period considered here, during which there were two clear periods and two mixed-phase cloudy periods. Data from a cloud imaging probe (CIP) at SPL (Figure 8) and visual verification were used to differentiate clear and cloud periods

One potential sampling concern was that wind-blown snow or ice fragments could enter the inlet and lead to incorrect ice crystal residual counts in SPIDER. Our data showed there was minimal / no association between wind speed, which was relatively low during this period, and ice crystal concentrations in clear conditions (see Supplement). The average wind speed during clear periods was 2.2 m/s (max: 5.5 m/s) for period 1 and 6 m/s (max: 8.3 m/s) for period 2 (see Supplement). Wind speeds ranging between 4-11 m/s have been shown to lead to fresh snow transport (Li and Pomeroy, 1997), but wind speed had little discernible influence on ice residual concentrations in SPIDER. Previous studies performed at SPL, such as Lowenthal et al. (2019), also found no association between ice crystal concentrations and wind speed. A background of small particles near the SP2-XR detection limit in the ice crystal residual channel not associated with wind was observed during clear conditions (Figures 9a and 9b). This inadvertent transmission is consistent with previous PCVI measurements (Pekour and Cziczo, 2011) and the lab studies and is eliminated from in cloud data.





During transitions into cloudy periods there was a change in the ice crystal residual concentrations comparable to the CIP and measurements of liquid water content (LWC) (Figure 8). Specifically, the ice crystal residual channel shows higher concentrations during the cloudy period, with an increase between 200 nm - 500 nm that is not observed during clear periods.

These results are consistent with previous studies using phase-separating inlets to measure ice residual concentration and size

in mixed-phase clouds (Mertes et al., 2007; Kupiszewski et al., 2016). Mertes et al. (2007) found that super-micrometer aerosol particles dominated ice residual concentrations whereas Kupiszewski et al. (2016) found ice residual particle sizes peaked at 200nm-300nm. Using an ice chamber to nucleate ambient aerosol, DeMott et al. (2010) found that INPs were typically >300 nm diameter. Our findings suggest an increase in ice residual concentration with increasing residual size from 200 to 500nm range. This should be considered in the context of the SP2-XR size range which was limited to 540 nm;

conclusions can therefore not be drawn about residual size into the super-micrometer diameter range.

## 6 Conclusion and Future Work

Laboratory studies have been used to show that both the individual components and composite SPIDER worked as designed. Field experiments were used to extend performance tests into mixed-phase clouds characterized by a suite of instruments at SPL.

In the laboratory verification measurements, each component of SPIDER was isolated and tested to validate that it performed its role in the overall system. Comparisons, where possible, were made with previous studies. These included the L-PCVI, the droplet evaporation chamber, and the PCVI. Once each component was verified individually, a test was done to ensure that the combination of instruments also functioned using either droplets or ice crystals. Future laboratory measurements should be conducted to further define the performance of both the L-PCVI and the droplet evaporation chamber. A full

characterization of the L-PCVI could better define the flow rates for obtaining an ideal D50 to segregate droplets and ice crystals from interstitial aerosol particle under a variety of ambient conditions. Similarly, it would be beneficial to perform more detailed laboratory experiments on droplet evaporation to determine performance in different cloud environments.

SPIDER was deployed at SPL in Colorado where it was tested during transitions from clear to cloud conditions. As a proof-of-concept campaign, SPIDER demonstrated that it was deployable to a remote field location. Sporadic failures in OPCs on

the interstitial and cloud drop residual channels led to minimal performance studies. Using a high-resolution SP2-XR, we were able to conclude minimal influence by windblown snow during this period and characterize inadvertent transmission similar to previous studies using PCVIs. Data from mixed-phase cloud events exhibited elevated counts of ice residuals that correlated with CIP and LWC measurements. The size of ice residuals, although limited by the detection of the SP2-XR, compared with results from previous field studies. Future field experiments with SPIDER should include more robust /

higher resolution OPCs on both the interstitial aerosol particle and cloud droplet residual channel. An OPC with a larger size range would be of use on the ice residual channel.





The goal for SPIDER was to develop a comprehensive system for segregating and then sampling cloud elements. Through the laboratory verification tests and field campaign, we demonstrate that SPIDER is capable of sorting the three components of mixed-phase clouds into distinct channels. Future goals include coupling SPIDER to particle mass spectrometers in order
to determine chemical composition. This would allow for a determination of the differences between particles that nucleate droplets or ice from unactivated particles. Ultimately, information on cloud nucleation capabilities of various aerosol particles could be compared to laboratory work and integrated into climate models (Shupe et al. 2008).

## 7 Acknowledgements

We would like to thank the William and Kumi Martin Foundation and the National Science Foundation (Grants AGS-
1749851 and AGS-1749865) for funding. L. Koolik would like to thank the MIT Department of Civil and Environmental Engineering for providing fellowship funding. We gratefully acknowledge Eric Kelsey and the staff of the Mount Washington Observatory for their assistance with the logistics associated an initial proof of concept study. We also thank the other members of the Cziczo Group at MIT for their assistance throughout the development of SPIDER. The authors appreciate the dedication, commitment, and effort of Maria Garcia and Dan Gilchrist towards the deployment of SPIDER at
SPL. We further thank Droplet Measurement Technology for the use of an SP2-XR for characterization of the ice residual channel. The Steamboat Ski Resort provided logistical support and in-kind donations. The Desert Research Institute is a permittee of the Medicine-Bow Routt National Forests and is an equal opportunity service provider and employer.

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



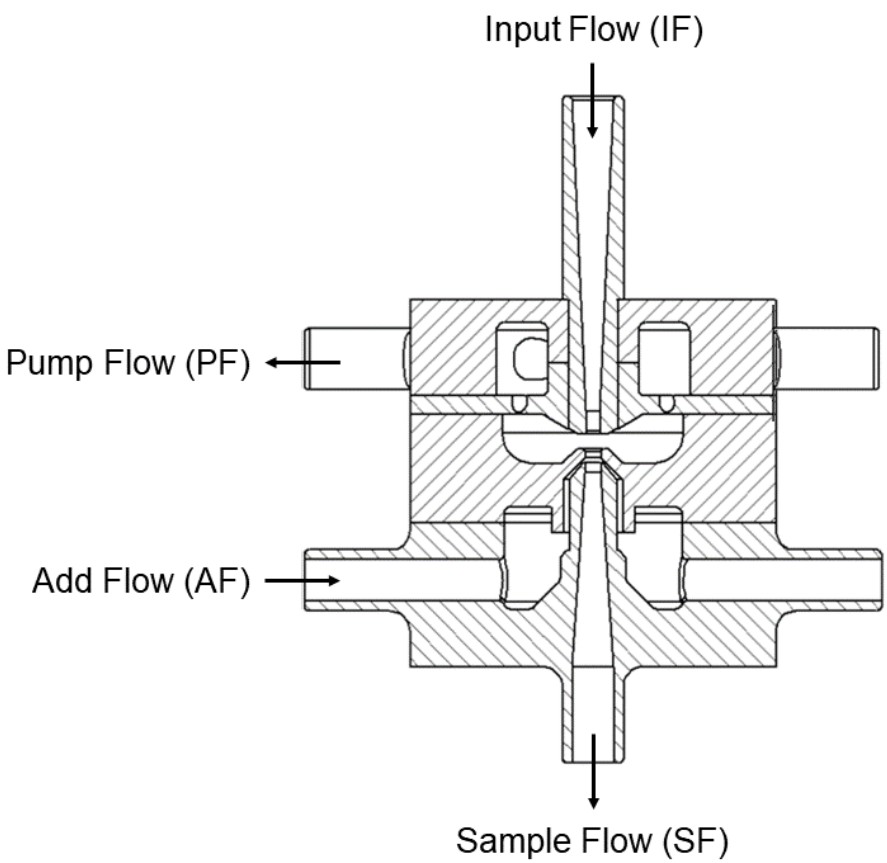


**Figure 1: Cross-sectional view of the 3D printed SPIDER PCVI with flows labelled. The 3D printed PCVI features the improved conical input nozzle suggested by Kulkarni et al. (2011); otherwise, the design is the same as considered by Kulkarni et al. (2011).**





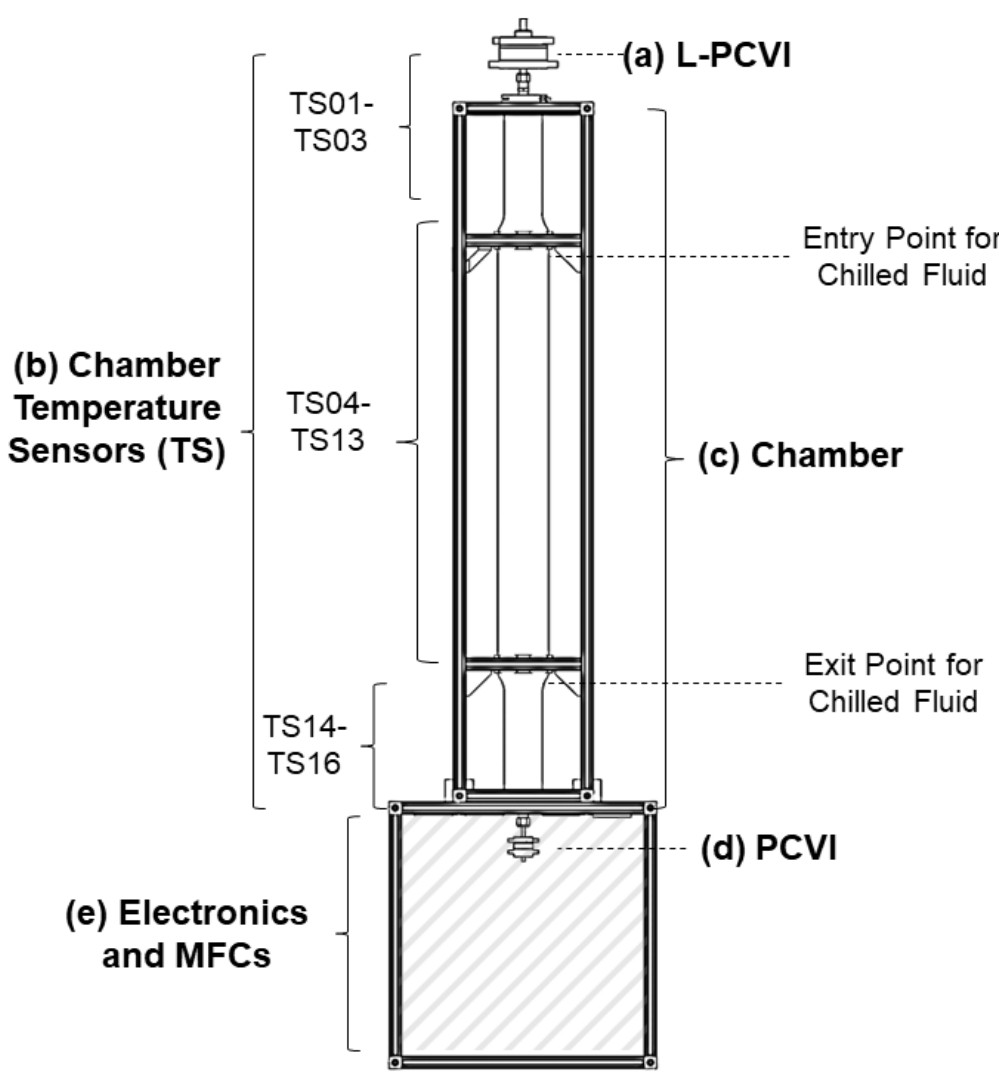





**Figure 2: Schematic of SPIDER with its components labelled. (a) The L-PCVI (Hiranuma et al., 2016) separates interstitial aerosol from the droplets and ice crystals. (b) Thermocouples report the temperature in the chamber. (c) The chamber is cooled and held at ice saturation to evaporate droplets. (d) The PCVI downstream separates evaporated droplet residuals from ice crystals. (e) The bottom houses electronics and mass flow controllers (MFCs).**

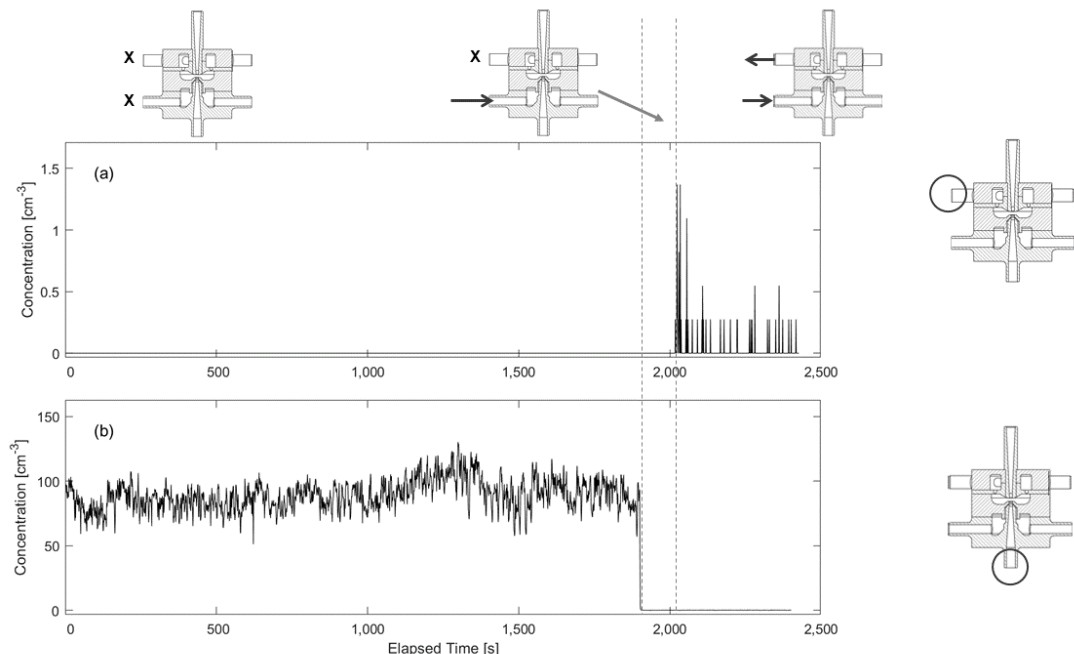

**Figure 3: Concurrent time series of particles detected in the (a) PF and the (b) SF of the L-PCVI. The flows into the PF and AF were turned on between 1,700-1,900 s. Note that particles were detected only in the SF until the flows were turned on, at which point the particles are counted in the PF. The "X" mark represents no flow through the corresponding chamber of the L-PCVI; an arrow represents flow. The circle indicates where the sample is measured.**





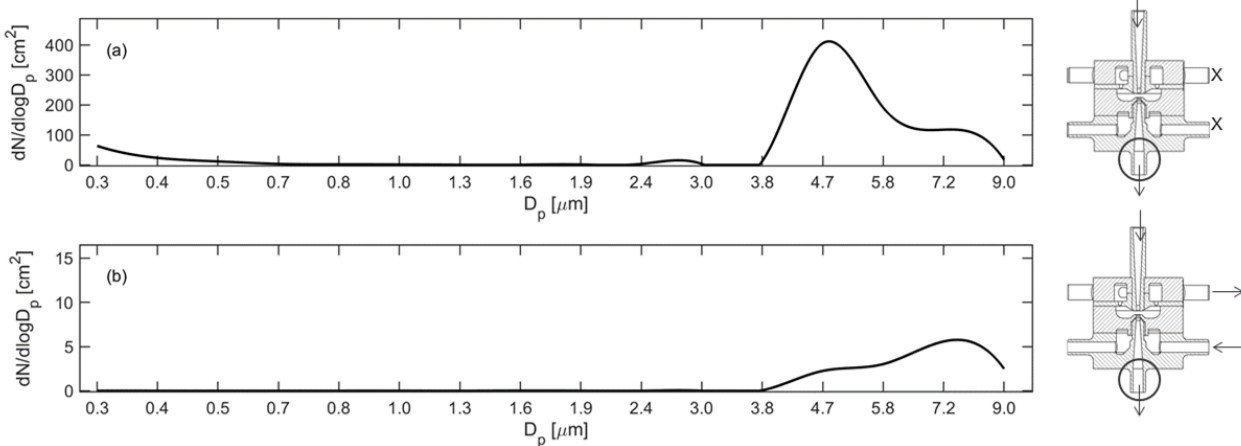

**Figure 4: Droplet transmission through the L-PCVI under varying flow scenarios. The average number concentration of particles out of the SF of the L-PCVI is plotted against particle diameter for (a) AF and PF off, (b) AF and PF on, and (c) only AF on.**

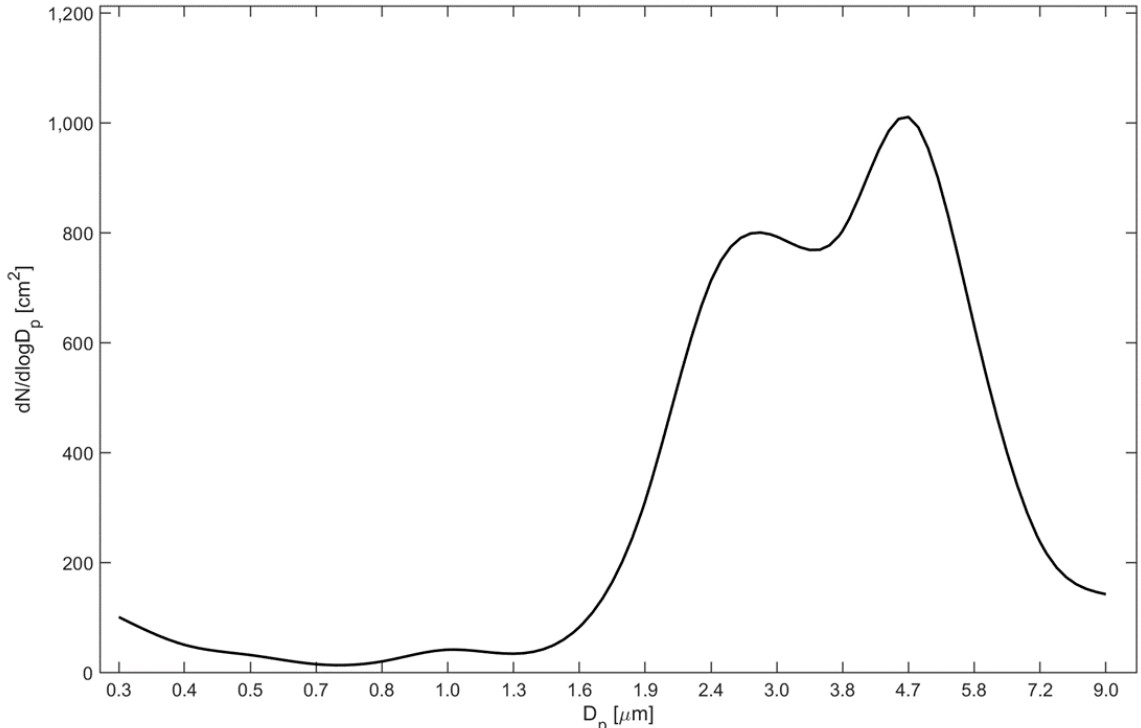

**Figure 5: Number concentration as a function of particle diameter for ice crystals travelling through the chamber. Ambient air was introduced to the cooled chamber with the PCVI on. Droplets were introduced into the cooled chamber with the PCVI on. The results shown are the difference between these two averages, reflecting the size distribution from ice crystals alone.**


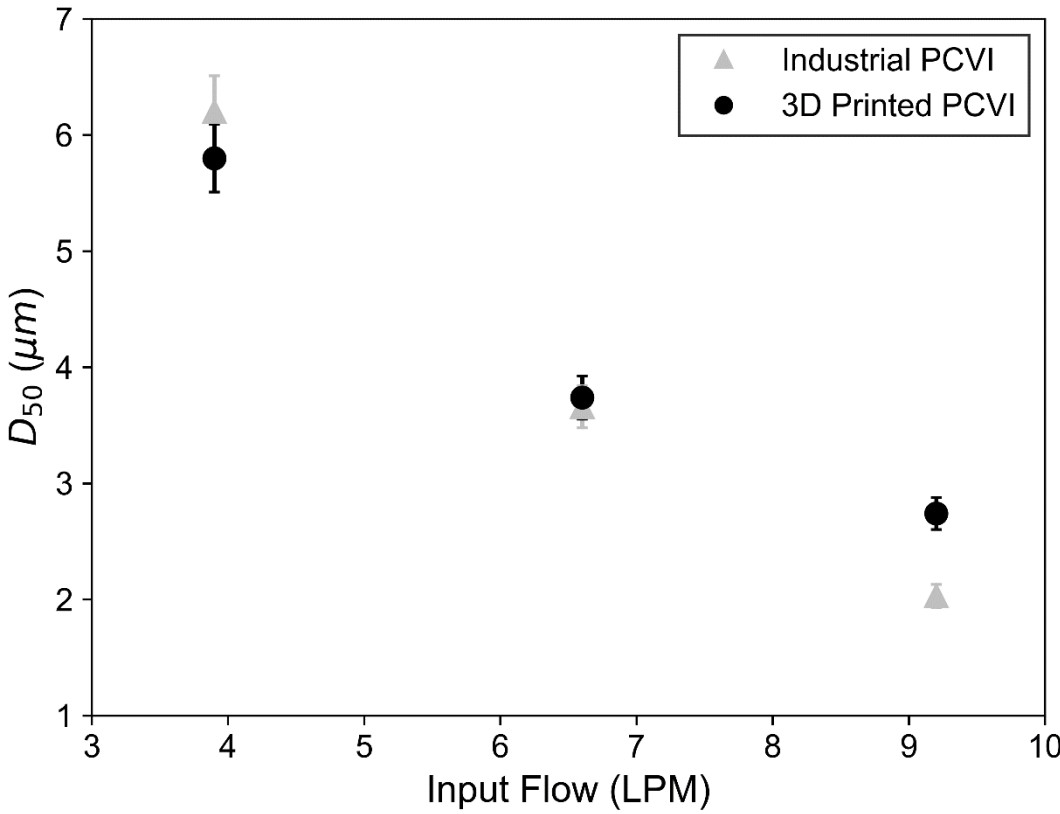

**Figure 6: Comparison of D50 values for the industrial and printed PCVI as a function of IF in the overlapping range of functionality.**





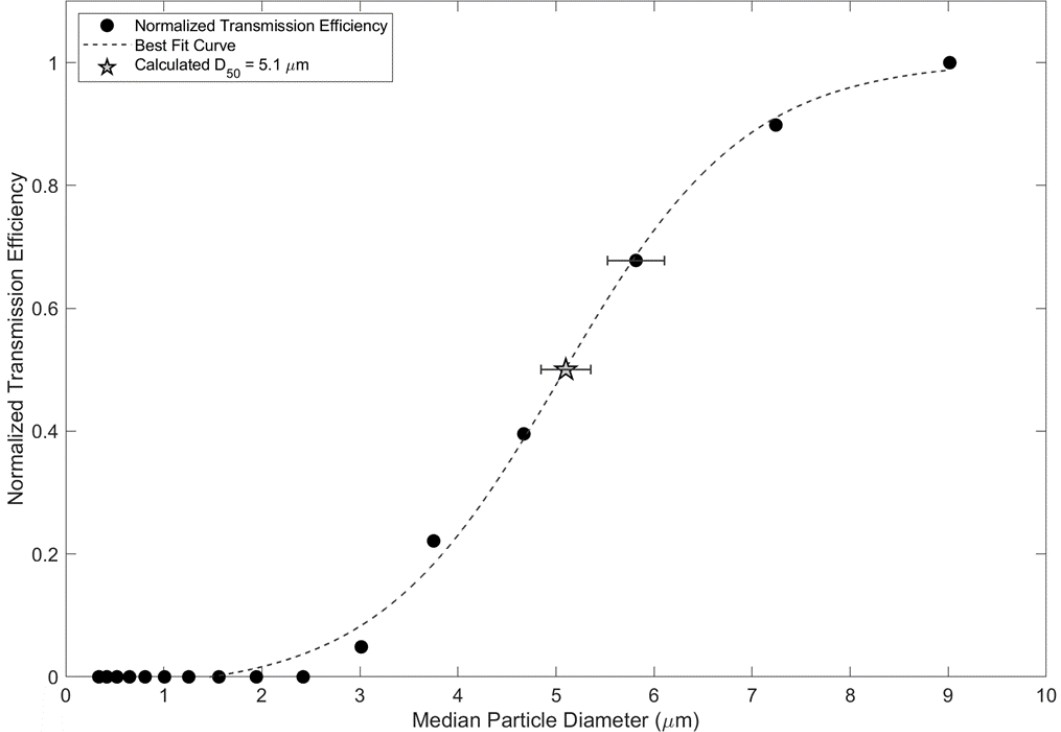

**Figure 7: Transmission efficiency as a function of particle diameter (solid circles) fit with a sigmoidal curve (dashed line). The star represents the size at which 50% of particles are transmitted (the experimental D50). The representative error, ±5% due to instrument uncertainty, is shown on a point close to the D50 and on the D50.**


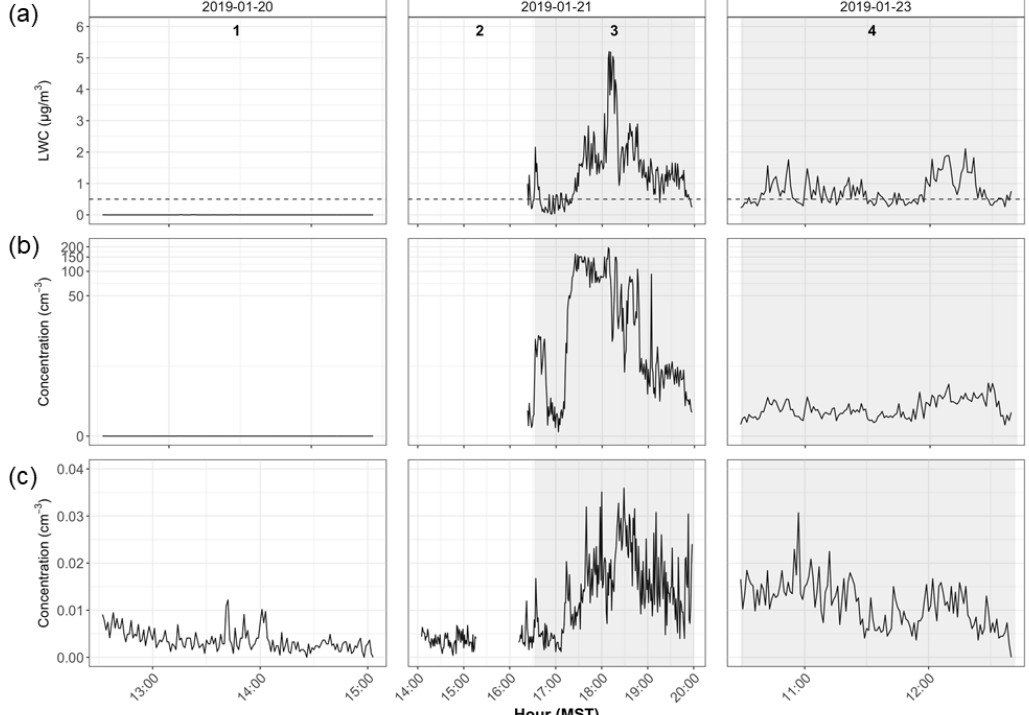

**Figure 8: (a) Cloud Imaging Probe water content over time for particles 30µm-105µm. Water content was calculated assuming spherical particles and a water density of 1 g/cm³. Dashed line at 0.025 µg/cm³ indicates the threshold for cloud presence. (b) Cloud Imaging Probe total number concentration over time where a logarithmic scale is used for clarity. (c)SP2-XR ice crystal residual concentration. Gray shading represents cloudy conditions.**






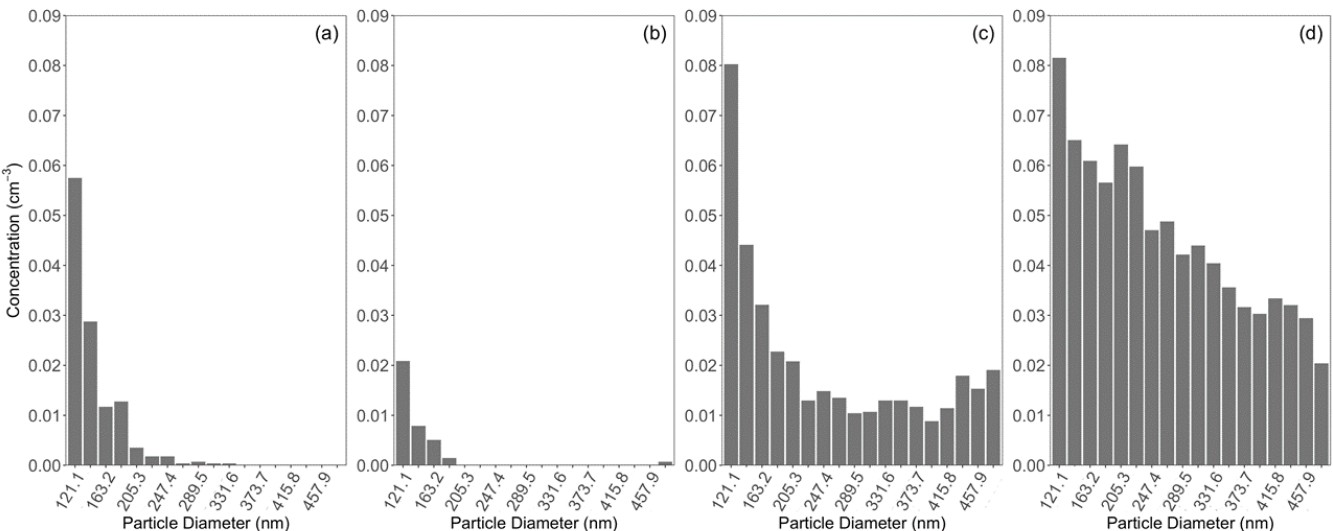

**Figure 9: SP2-XR average ice crystal residual concentration. Clear conditions during periods 1 and 2 (a, b) and cloudy conditions during periods 3 and 4 (c, d). Average background concentrations during clear conditions were subtracted from cloudy conditions.**

**Table 1: L-PCVI Flow Tests**

| AF-to-IF Ratio | Flows (L min⁻¹) | | | |
|---|---|---|---|---|
| | **AF** | **IF** | **PF** | **SF** |
| 0.14* | 7.0 | 50.0 | | 2.0 |
| 0.15* | 11.5 | 75.0 | | 2.5-6.0 |
| 0.16* | 11.5 | 70.0 | | 2.5-6.0 |
| 0.14** | 7.0 | 50.0 | 55.0 | 2.0 |
| 0.16** | 11.5 | 72.7 | 77.7 | 6.5 |
| 0.23** | 11.5 | 50.0 | 55.0 | 6.5 |

*Ratios and flows used by Hiranuma et al. (2016), the PF was not provided by the authors as their SF was varied. **Ratios and flows used in this study.