# Peer review of "A Phase Separation Inlet for Droplets, Ice Residuals, and Interstitial Aerosol Particles"

_Atmospheric Measurement Techniques, 2021_

## Referee Comment (RC2)

AUTHORS:

KOOLIK, ROESCH, DELOOYA, SHEN, HALLAR, MCCUBBIN, CZICZO

Manuscript title:

A phase separation inlet for droplets, ice residuals, and interstitial aerosol particles

**General evaluation:**

The idea is definitely ambitious to design and build such a complicated inlet. However, the manuscript title is more ambitious than the actual instrument performance that is lacking an acceptable quantitative assessment of limitations and possible artefacts like particle contamination, particularly from larger droplets.

As it stands, proof of a proper operation of the compete SPIDER instrument are very limited, due to incomplete residual particle measurements and also lack of necessary complementary data in the field as for example droplet spectrum. The 'Conclusion' section of the manuscript illustrates the lack of proper work (lines 273-277), talking of minimal performance studies (also due to instrument failures). I would rather say 'insufficient performance studies'!

At this stage the performed work is something like a proof of concept of phase separation, but with significant limitations. Those limitations have to be either revised (which would be best) or at least quantified (stating limited operability of SPIDER instrument), and this is why the manuscript needs major revision.

**Major concerns and comments to be addressed:**

- I understood that the droplet evaporation chamber upper limit is 20 µm which is a considerable problem. I guess the concentration of D>20 µm atmospheric supercooled droplets at SPL is of the same order of magnitude as is the crystal concentration. Consequently, crystal residual concentration in the PCVI is a major problem, since SPL is in a rather clean environment with important MVDs? What's the consequence of that? Droplet residual particles more or less dominating the ice residual size distribution in Fig 9? Thus, the SPL campaign is of no use, when you don't know the supercooled droplet spectrum! Looking into droplet spectra of past SPL measurements, you definitely have non negligible numbers of droplets beyond 20µm in diameter.

- Figures 4a) and 4b): 50µm / 40 µm droplets are difficult to handle. What is the effect of droplet breakup on both figures? It seems you applied a multimodal fit to measurements? Why? Please explain what's happening in Fig 4a and 4b? If we are concentrating on 'modes' 4.7 and 7.2 µm, why is the size distribution so different? 4.7 µm mode dominates Fig 4a and 7.2µm Fig 4b, why?

- Figure 4 c) is missing.

- Line 181… experiments bracket D50 of L-PCVI between 10-40 µm needs to be explained. Under which 'flow' conditions 10µm and under which 40 µm?

- Figure 5: Why do we see essentially a bimodal distribution, if this is not an artefact?

- Figure 7: figure caption "The representative error, +/-5%; due to instrument uncertainty,… . What do you call a representative error? Pease give an equation how the error is defined and quantify what is meant with instrument uncertainty!

- Figure 8a): µg/m3 is certainly a false unit. Probably mg/m3 would be also false, don't think that a LWC-300 can resolve 1mg of supercooled water? Please clarify!
- In addition I'd like to see the LWC signal in clear sky before 16:45
-
- Figure 8b): Likewise this figure is not comprehensive. I've never seen CIP concentrations of 150/ccm. Never seen a drizzle or crystal concentration of that magnitude. Impossible! If it is droplets, the PCVI must be completely contaminated with droplet residuals… and likewise this can't be crystals of that concentration. This is simply impossible from microphysics.
-
- Figure 8b): Likewise please show CIP signal before 16:45 in clear sky. What is your confidence in the CIP concentrations? And 30 µm particle size seems to represent 2 pixel? Droplet or crystal? Concentration problem see above!
-
- Figures 8b) and 8c): Another major concern is the comparison of periods 3 & 4: The ice residual concentrations in periods 3 and 4 are comparable (factor of 2 and closer), however the crystal concentrations are off by a factor of 50? I wouldn't expect that, and you have to explain the lack of measurement coherence. I thought one ice crystal releases one crystal residual. As explained above I wouldn't expect 100-150 drops of D>30µm. Those would all end up in the PCVI….

- Is secondary ice production at SPL a subject to be considered? What are the consequences for SPIDER data interpretation, when secondary ice exists?

Minor comments:
- Line 64 bracket missing
- Line 88 'with higher tolerance' explanation and quantification of what this means
- Line 96: IS-PCVI?
- Lines 118-134: Suggest that 3D printing details are not necessary here
- Line 136ff: Why didn't you install a simple CPC counter to prove absence of (i) interstitial aerosol transmission into L-PCVI and (ii) droplet residual transmission into PCVI?
- In order to detect small particle contamination (interstitial aerosol going through L-PCVI; droplets and / or drop residuals going through the PCVI) you may just use a CPC counter to exclude contamination. As presented, you can't rule out that

possibility. Is there any reason not to verify just particle concentrations of 0, instead of just looking at accumulation mode with OPCs?

- Line 151-152: characterization in the style of Boulter et al not possible. Leave this out or explain the characterization method.
- Line 152-153 sentence not clear. Explain
- Line 197: guess AS means ammonium sulphate
- Line 222-223: SPL contains a measurement suite for aerosol particles, cloud properties…. Which cloud properties and related instruments in addition to CIP imager to claim SPL an ideal site for SPIDER deployment. As it stands (this manuscript) you only had SPIDER plus SP2-XR plus CIP at the site. Everything else but ideal? OPCs failed, no CPCc, no complementary droplet spectrum? Also SP2-XR is measuring accumulation mode black carbon mass and size and not whatever contamination from interstitial Aitken and non-carbon particles and/or droplet residuals.

---

## Author Comment (AC1)

We would like to start this response to Reviewer document with a thank you to all three Reviewers for their work to help us improve the manuscript. We also thank your patience; the length of time between review and revision was that we took your comments very seriously. First, we have conducted a new and comprehensive set of experiments to validate all inlet components as well as the composite system. This was the source of many review points from all three Reviewers. Second, we have removed the Storm Peak experiments, the source of several comments. We believe this streamlines and focuses the paper on the inlet calibration work. Specific changes are outlined below in a point by point format, including reference improvement, with our response in *italics*. With regard to the extensive new experiments we will often refer to 'please see new content for details' for simplicity.

**Responses to Reviewer 1**

Transmission efficiencies: As you claim that the SPIDER inlet is able to sample simultaneously interstitial aerosol particle, droplet residuals, and ice crystal residuals, it would be needed to address the transmission efficiencies for the different channels. To my understanding this can be retrieved from the existing measurements. No particles smaller than the lower size limits of the OPC/OPS (~0.3 μm) were measured. Hiranuma et al. (2016) used a condensation particle counter to address the question of transmission of small particles in the different channels. In my opinion, such measurements can help to verify that e.g. no small aerosol particles or small evaporated cloud droplets are able to be transmitted in the droplet or ice channel, respectively. Further, such measurements can also be used for transmission efficiency measurements at the interstitial aerosol channel. This is rather a recommendation for future work and does not imply that new measurements need to be presented in this manuscript.

*We agree with the reviewer that the paper needed an increased level of detail on transmission efficiency and this is now incorporated into the manuscript. We clarify (as added in the paper per this and following comments by all Reviewers) that any CVI is subject to inadvertent transmission. Thus, a CPC will not show a 0 counts but instead reflect the rate of inadvertent transmission. As the Reviewer points out, the OPC measurements detailed here are meant to determine this rate for the size range of interest.*

Ice crystal residuals and cloud droplet residuals are not necessarily only INPs and CCNs, respectively, as cloud droplets can also contain scavenged particles and ice residuals can also contain droplet residuals due impact from secondary ice crystal formation (e.g. see discussion in Kamphus et al., 2010). Based on your statement in the introduction (lines 67 – 70) and in the conclusions (lines 290 – 291), you should be more specific about what ice residuals and cloud droplet residuals are when you sample them with SPIDER. Are you truly only measuring INPs and CCNs?

*The reviewer raises an important point. While we can not determine, a priori, these processes, we now note in the text that there "is not necessarily a 1 to 1 relationship between droplets and ice crystals and residuals. Droplets or ice crystals can scavenge gas- and particle-phase constituents. Droplets and ice crystals can also undergo breakup (more detail in the following sections) or secondary formation processes. The purpose of this work is to detail a means for separation of interstitial aerosol, droplets and ice crystals into three separate channels. The specific cloud properties, such as cloud lifetime, scavenging rates, breakup processes and secondary hydrometeor production mechanisms, at a sampling site will dictate the efficacy of SPIDER to resolve residuals. "*

Specific comments:

Abstract: I suggest to give the size range of ice particles which can be analyzed with SPIDER.

*The size range of ice particles that can be analyzed with SPIDER (2.7 - 25 µm) has been included in the abstract.*

Lines 26 – 28: I assume that the most important criteria about the Storm Peak Laboratory campaign was that you were able to sample ambient supercooled clouds, which I would mention here.

*Now removed from text*

Lines 29 – 30: „Possible design improvements of SPIDER are also suggested", are you refering here to using more robust OPCs or OPCs with a higher resolution? It is not clear to me what those design improvements would be.

*See new information with APS and OPS for calibration*

Lines 33 – 34: „Mixed-phase clouds are important factors in aviation and climate (Shupe et al., 2008)", please add more and also more recent literature, as e.g. Lohmann et al. (2017), McCoy et al. (2016).

*We have added references to McCoy et al. (2016) and Lohmann (2017).*

Lines 38 – 40: „Mixed-phase clouds are particularly complicated because the partitioning of phases is critical in assessing these effects (Atkinson et al., 2013; Hirst et al., 2001; Korolev et al., 2003; Shupe et al., 2006)." Atkinson et al. (2013) was not investigating this specific research question; also, there is more and also more recent literature about this, e.g. Korolev et al. (2017), Tan and Storelvmo et al. (2019), just to name a few.

*We have replaced Atkinson et al. (2013) with the suggested literature.*

Lines 41 – 43: „This has resulted in a global effort to study these clouds (Abel et al., 2014; Davis et al., 2007a; Hiranuma et al., 2016; Kupiszewski et al., 2015; Mertes et al., 2007; Patade et al., 2016)." Also here, include more recent studies, e.g. Lohmann et al. (2017), Lowenthal et al. (2019), Schmidt et al. (2017), Ramelli et al. (2021), Ruiz-Donoso et al. (2020).

*The suggested literature has been added.*

Lines 45 – 46: „At this saturation aqueous droplets are the favored state and particles that activate are termed cloud condensation nuclei (CCN) (Lohmann and Hoose, 2009; Wang et al., 2012)." Those references are not specifically investigating warm cloud actication. I recommend to change the references to e.g. Pruppacher and Klett (1997).

*We have replaced the previous literature referenced with Pruppacher and Klett (1997) as suggested.*

Lines 42 – 49: „Ice can form homogeneously, via spontaneous nucleation of ice in a solution droplet, at temperatures below -40°C (Atkinson et al., 2013; Kamphus et al., 2010; Korolev et al., 2003; Storelvmo et al., 2008; Verheggen et al., 2007; Wang et al., 2012)." None of those publications focus on homogeneous freezing of solution droplets, I recommend to reference Heymsfield et al. (2017) or Koop et al. (2000).

*We have replaced the previous literature referenced with the suggested references*

Lines 49 – 51: „At higher temperatures, ice forms heterogeneously through different pathways promoted by ice nucleating particles (INPs) (Atkinson et al., 2013; Kamphus et al., 2010; Lohmann and Hoose, 2009; Storelvmo et al., 2008; Tsushima et al., 2006; Verheggen et al., 2007; Wang et al., 2012)." I recommend to reference rather review papers specifically on INPs, e.g. Hoose and Möhler (2012), Kanji et al. (2017).

*We have replaced the previous literature referenced with the suggested references.*

Lines 51 – 52: „The specific properties that determine an effective INP remain poorly understood (Shupe et al., 2008)." Shupe et al. (2008) did not investigate INP properties. I suggest to reference Kanji et al. (2017).

*We have replaced the previous literature referenced with the suggested references.*

Lines 62 – 63: „Motivated by climate change, estimated to be warming approximately twice as fast as the global average (Verlinde et al., 2007)…" please reference more recent literature here.

*Revised this section and replaced with more recent Arctic INP literature. Additionally, more recent literature has been added for INP research at Jungfraujoch.*

Line 94: Please introduce the abbreviation for IS-PCVI.

*Updated to include the full name of the IS-PCVI.*

Lines 102 – 103: I suggest to include the expected D50 for those flow settings.

*We have now included the expected D50 range from the Hiranuma et al., 2016 paper, as well as a reference to the section where we determine the D50 experimentally in this work. Please see also new work to better define the D50.*

Line 103: I suggest to move „the PCVI PF, AF, and SF at 8.0, 2.5, and 1.0 L min-1, respectively." to below when you introduce the PCVI, e.g. to lines 111 – 112.

*We have moved the PCVI flow conditions as suggested.*

Lines 103 – 105: I suggest to give the Weber Number here (0.3) in comparison to a value of 10 and larger when droplet breakup is expected.

*The Weber number and threshold are included for the L-PCVI and PCVI.*

Section L-PCVI: Based on the experiments presented in Fig. 3 and 4 you could determine the transmission efficiency of interstitial particles in the PF, taking into account the dilution ratio.

*While we agree this type of experiment was possible it was beyond the scope of the experiments we conducted. Previous studies suggest this as ~0.8 which we default to without experimental evidence to the contrary.*

Section Droplet Evaporation Chamber: What is the residence time of cloud droplets and ice crystals in the droplet evaporation section, and does this impact the partial evaporation?

*The residence time (~25 s) and additional data on evaporation has been added.*

Section Sustaining Ice Crystals: It is not clear to me which „chamber" is meant here. Was the droplet evaporation chamber used to induce homogeneous freezing and form ice crystals? If so, how could you determine if ice crystals survived in the droplet evaporation chamber? As this section belongs to 4.2, I understand that the intention is to test if ice crystals are sustaining in the droplet evaporation chamber, which, in theory, is not needed, as the droplet evaporation chamber is maintained at ice-coated walls (saturated with respect to ice). Maybe you should consider to move this section to 4.3.

*For clarity, this is now stated as the 'evaporation chamber'. We have rewritten this section and now include quantitative evaporation and ice crystal using a new methodology with ice formed above the chamber. We do believe the evaporation of droplets and maintenance of ice, the two requirements of the evaporation chamber, are now more clearly explained and separated in the text.*

Lines 198 - 199: What is the size of the formed ice crystals? And how were they validated visually?

*See above, now with direct detection using OPS and APS.*

Line 206: Which particle sizes are generated with this AS solution?

*We have added a figure to the Supplement showing the size distribution of AS particles generated by the bubbler.*

Lines 234 – 235: I recommend to not include the dicsussion about the different OPCs used at the interstitial aerosol channel and cloud residual channel, as you don´t show those results.

*This section is now removed and replaced with the new instrument calibration tests.*

Line 241: I suggest to give a number for „low aerosol particle conditions".

*We have now included a range of particle number concentration for background.*

Lines 249 – 250: This is a repetition from your statement in line 246, I would delete one of the sentences.

*We have removed the initial statement*

Lines 253 – 254: Where is this „inadvertent transmission" coming from?
Updated to describe the source of inadvertent transmission.        Lines 290 – 291: „Ultimately, information on cloud nucleation capabilities of various
aerosol particles could be compared to laboratory work and integrated into climate models (Shupe et al. 2008)" I recommend to cite also more recent literature here.

*We have added two recent citations.*

The author contributions is missing

*Added the author contribution statement, as well as the competing interests statement.*

Figure 4: There is no panel (c)

*Figure 4C was removed in a previous iteration. All references to old Figure 4C have now been removed.*

Figure 7: Please indicate that this is the transmission efficiency from the PCVI

*Updated to indicate which is L-PCVI and PCVI.*

Figure 8 (and related discussion in the text): Another important parameter for the description of these timeseries would be the ambient temperature, which one could relate to the nucleation temperature of ice crystals in the cloud. More, on 2019-01-21 at ~ 18:30, ice crystal concentrations are as high as 0.03 cm-3, which is a relative high INP concentration at temperatuers < -20°C (the lower limit of ambient temperature, as

I understand from line 240). Thus, is this an indication for an impact of sampling ice crystals formed by secondary processes?

*Note that in response to comments by all three reviewers we decided to withhold review until after more comprehensive experiments were conducted and also eliminated the Storm Peak data for clarity.*

Figure 9 (panels c, d): Also here, are your measurements impacted by secondary ice crystal production in the smaller size bins? It is quite surprising that the concentration of ice crystal residuals increase towards smaller sizes.

*Now eliminated from the paper.*

Editorial comments:

Line 92: Please introduce the abbreviation „SPIDER", since it is the first time using it in the main text.

*The full name of SPIDER is now included.*

Line 93: Please introduce the abbreviation „L-PCVI".

*The full name of the L-PCVI is now included at the first reference.*

Line 222: The abbreviation for INP was introduced earlier.

*The full name of CCN and INP are removed.*

Lines 304 – 305: Remove those test citations.

*The test citations have been removed.*

I suggest to either use supersaturation or relative humidity with respect to water (especially in the droplet evaporation chamber section).
*Updated to (super)saturation.*

As the abbreviation for ammonium sulfate (AS) is only used a few times in the manuscript I suggest to use the full name.

*We have updated to change "AS" to "ammonium sulfate" globally.*

The resolution of Figures 3, 4, 8, and 9 can be improved.

*Figure resolution has been updated globally.*

References

Heymsfield, A. J., Krämer, M., Luebke, A., Brown, P., Cziczo, D. J., Franklin, C., Lawson, P., Lohmann, U., McFarquhar, G., Ulanowski, Z., and Van Tricht, K.: Cirrus Clouds, Meteorological Monographs, 58, 2.1-2.26, 10.1175/amsmonographs-d-16-0010.1, 2017.

Hoose, C., and Möhler, O.: Heterogeneous ice nucleation on atmospheric aerosols: a review of results from laboratory experiments, Atmos. Chem. Phys., 12, 9817 - 9854, 10.5194/acp-12-9817-2012, 2012.

Kanji, Z. A., Ladino, L. A., Wex, H., Boose, Y., Burkert-Kohn, M., Cziczo, D. J., and Krämer, M.: Overview of Ice Nucleating Particles, Meteorological Monograophs, 58, 1.1-1.33, 10.1175/amsmonographs-d-16-0006.1, 2017.

Koop, T., Luo, B., Tsias, A., and Peter, T.: Water activity as the determinant for homogeneous ice nucleation in aqueous solutions, Nature, 406, 611-614, 10.1038/35020537, 2000a.

Korolev, A., McFarquhar, G., Field, P. R., Franklin, C., Lawson, P., Wang, Z., Williams, E., Abel, S. J., Axisa, D., Borrmann, S., Crosier, J., Fugal, J., Krämer, M., Lohmann, U., Schlenczek, O., Schnaiter, M., and Wendisch, M.: Mixed-Phase Clouds: Progress and Challenges, Meteorol. Monogr., 58, 5.1-5.50, 10.1175/amsmonographs-d-17-0001.1, 2017.

Lohmann, U.: Anthropogenic Aerosol Influences on Mixed-Phase Clouds, Current Climate Change Reports, 3, 32-44, 10.1007/s40641-017-0059-9, 2017.

Lowenthal, D. H., Hallar, A. G., David, R. O., McCubbin, I. B., Borys, R. D., and Mace, G. G.: Mixed-phase orographic cloud microphysics during StormVEx and IFRACS, Atmos. Chem. Phys., 19, 5387-5401, 10.5194/acp-19-5387-2019, 2019.

McCoy, D. T., Tan, I., Hartmann, D. L., Zelinka, M. D., and Storelvmo, T.: On the relationships among cloud cover, mixed-phase partitioning, and planetary albedo in GCMs, Journal of Advances in Modeling Earth Systems, 8, 650-668, https://doi.org/10.1002/2015MS000589, 2016.

Pruppacher, H. R., and Klett, J. D.: Microphysics of Clouds and Precipitation, Kluwer Acad. Norwell, Mass, 1997.

Ramelli, F., Henneberger, J., David, R. O., Lauber, A., Pasquier, J. T., Wieder, J., Bühl, J., Seifert, P., Engelmann, R., Hervo, M., and Lohmann, U.: Influence of low-level blocking and turbulence on the microphysics of a mixed-phase cloud in an inner-Alpine valley, Atmos. Chem. Phys., 21, 5151-5172, 10.5194/acp-21-5151-2021, 2021.

Ruiz-Donoso, E., Ehrlich, A., Schäfer, M., Jäkel, E., Schemann, V., Crewell, S., Mech, M., Kulla, B. S., Kliesch, L. L., Neuber, R., and Wendisch, M.: Small-scale structure of thermodynamic phase in Arctic mixed-phase clouds observed by airborne remote sensing during a cold air outbreak and a warm air advection event, Atmos. Chem. Phys., 20, 5487-5511, 10.5194/acp-20-5487-2020, 2020.

Schmidt, S., Schneider, J., Klimach, T., Mertes, S., Schenk, L. P., Kupiszewski, P., Curtius, J., and Borrmann, S.: Online single particle analysis of ice particle residuals from mountain-top mixed-phase clouds using laboratory derived particle type assignment, Atmos. Chem. Phys., 17, 575-594, 10.5194/acp-17-575-2017, 2017.

Tan, I., and Storelvmo, T.: Evidence of Strong Contributions From Mixed-Phase Clouds to Arctic Climate Change, Geophysical Research Letters, 46, 2894-2902, https://doi.org/10.1029/2018GL081871, 2019.

---

## Author Comment (AC2)

We would like to start this response to Reviewer document with a thank you to all three Reviewers for their work to help us improve the manuscript. We also thank your patience; the length of time between review and revision was that we took your comments very seriously. First, we have conducted a new and comprehensive set of experiments to validate all inlet components as well as the composite system. This was the source of many review points from all three Reviewers. Second, we have removed the Storm Peak experiments, the source of several comments. We believe this streamlines and focuses the paper on the inlet calibration work. Specific changes are outlined below in a point by point format, including reference improvement, with our response in *italics*. With regard to the extensive new experiments we will often refer to 'please see new content for details' for simplicity.

**Responses to Reviewer 2**

Major concerns and comments to be addressed:

- I understood that the droplet evaporation chamber upper limit is 20 µm which is a considerable problem. I guess the concentration of D>20 µm atmospheric supercooled droplets at SPL is of the same order of magnitude as is the crystal concentration. Consequently, crystal residual concentration in the PCVI is a major problem, since SPL is in a rather clean environment with important MVDs? What's the consequence of that? Droplet residual particles more or less dominating the ice residual size distribution in Fig 9? Thus, the SPL campaign is of no use, when you don't know the supercooled droplet spectrum! Looking into droplet spectra of past SPL measurements, you definitely have non negligible numbers of droplets beyond 20µm in diameter.

*We have rewritten this section to avoid confusion that we inadvertently caused. In brief, the SPIDER L-PCVI cut size is the lower hydrometeor bound whereas the upper D50 is set by the facility inlet system. Hydrometeors of unlimited size are rejected at this stage. To clarify we add "The cut size L-PCVI sets the lower size limit of droplets and/or ice transmitted into the SF. The upper cut size is set by the inlet from which SPIDER is sampling. In the case of the studies detailed in the following sections, the facility inlet at the Desert Research Institute's Storm Peak Laboratory (SPL), described by Petersen et al. (2019), was used, setting an upper D50 of 13 micrometers aerodynamic diameter and 25% particle transmission extending to 15 micrometers. " We believe this clarifies that droplets <25 micrometers are not actually input to SPIDER due to the facility inlet.*

- Figures 4a) and 4b): 50µm / 40 µm droplets are difficult to handle. What is the effect of droplet breakup on both figures? It seems you applied a multimodal fit to measurements? Why? Please explain what's happening in Fig 4a and 4b? If we are concentrating on 'modes' 4.7 and 7.2 µm, why is the size distribution so different? 4.7 µm mode dominates Fig 4a and 7.2µm Fig 4b, why?

*This section is now rewritten with the new calibration data. Please note the Weber number calculations regarding breakup.*

- Figure 4 c) is missing.

*Reference to Figure 4c has been removed globally.*

- Line 181… experiments bracket D50 of L-PCVI between 10-40 µm needs to be explained. Under which 'flow' conditions 10µm and under which 40 µm?

*The L-PCVI section has been extensively modified with new data which we believe clarifies flow conditions used in each experiment.*

- Figure 5: Why do we see essentially a bimodal distribution, if this is not an artefact?

*This section is now rewritten with the new calibration data.*

- Figure 7: figure caption "The representative error, +/-5%; due to instrument uncertainty,… . What do you call a representative error? Pease give an equation how the error is defined and quantify what is meant with instrument uncertainty!

*The representative error is provided by the instrument manufacturer. This is clarified in the updated figure caption.*

- Figure 8a): µg/m3 is certainly a false unit. Probably mg/m3 would be also false, don't think that a LWC-300 can resolve 1mg of supercooled water? Please clarify!

- In addition I'd like to see the LWC signal in clear sky before 16:45

- Figure 8b): Likewise this figure is not comprehensive. I've never seen CIP concentrations of 150/ccm. Never seen a drizzle or crystal concentration of that magnitude. Impossible! If it is droplets, the PCVI must be completely contaminated with droplet residuals… and likewise this can't be crystals of that concentration. This is simply impossible from microphysics.

- Figure 8b): Likewise please show CIP signal before 16:45 in clear sky. What is your confidence in the CIP concentrations? And 30 µm particle size seems to represent 2 pixel? Droplet or crystal? Concentration problem see above!

- Figures 8b) and 8c): Another major concern is the comparison of periods 3 & 4: The ice residual concentrations in periods 3 and 4 are comparable (factor of 2 and closer), however the crystal concentrations are off by a factor of 50? I wouldn't expect that, and you have to explain the lack of measurement coherence. I thought one ice crystal releases one crystal residual. As explained above I wouldn't expect 100-150 drops of D>30µm. Those would all end up in the PCVI….

- Is secondary ice production at SPL a subject to be considered? What are the consequences for SPIDER data interpretation, when secondary ice exists?

*Storm Peak data, based on these and other comments, have been eliminated in favor of a more extensive set of laboratory validation and calibration experiments.*

Minor comments:
Secondary ice production would lead to crystals without a residual particle in our detection range. Crystals would be counted by facility cloud probes but not the SPIDER instrumentation.

*Storm Peak data, based on these and other comments, have been eliminated in favor of a more extensive set of laboratory validation and calibration experiments.*

- Line 64 bracket missing

*The missing bracket has been added.*

- Line 88 'with higher tolerance' explanation and quantification of what this means

*We have removed the reference to higher tolerance, as the 3D printing technical details are outside the scope of this manuscript.*

- Line 96: IS-PCVI?

*The full name of the IS-PCVI has been added to the text.*

- Lines 118-134: Suggest that 3D printing details are not necessary here

*Updated to remove the description of 3D printing details.*

- Line 136ff: Why didn't you install a simple CPC counter to prove absence of (i)
interstitial aerosol transmission into L-PCVI and (ii) droplet residual transmission into
PCVI?

- In order to detect small particle contamination (interstitial aerosol going through LPCVI;
droplets and / or drop residuals going through the PCVI) you may just use a
CPC counter to exclude contamination. As presented, you can't rule out that
possibility. Is there any reason not to verify just particle concentrations of 0, instead of
just looking at accumulation mode with OPCs?

*We are unclear of the point of these comments, please clarify if necessary? The manuscript details inadvertent transmission, which has also been described in detail previously for both the PCVI and L-PCVI. We do not attempt to rule out this possibility. There should be a small component of inadvertent transmission, not an absence. The use of a CPC would show this inadvertent transmission but not be in the range of interest for these studies.*

- Line 151-152: characterization in the style of Boulter et al not possible. Leave this out
or explain the characterization method.

*Removed reference to the Boulter characterization in this section.*

- Line 152-153 sentence not clear. Explain

*Updated to clarify sentence.*

- Line 197: guess AS means ammonium sulphate

*We have updated to change "AS" to "ammonium sulfate" globally.*

- Line 222-223: SPL contains a measurement suite for aerosol particles, cloud properties…. Which cloud properties and related instruments in addition to CIP imager to claim SPL an ideal site for SPIDER deployment. As it stands (this manuscript) you only had SPIDER plus SP2-XR plus CIP at the site. Everything else but ideal? OPCs failed, no CPCc, no complementary droplet spectrum? Also SP2-XR is measuring accumulation mode black carbon mass and size and not whatever contamination from interstitial Aitken and non-carbon particles and/or droplet residuals.

*Please see above comments on removal of SPL data for clarity.*

---

## Author Comment (AC4)

We would like to start this response to Reviewer document with a thank you to all three Reviewers for their work to help us improve the manuscript. We also thank your patience; the length of time between review and revision was that we took your comments very seriously. First, we have conducted a new and comprehensive set of experiments to validate all inlet components as well as the composite system. This was the source of many review points from all three Reviewers. Second, we have removed the Storm Peak experiments, the source of several comments. We believe this streamlines and focuses the paper on the inlet calibration work. Specific changes are outlined below in a point by point format, including reference improvement, with our response in *italics*. With regard to the extensive new experiments we will often refer to 'please see new content for details' for simplicity.

**Responses to Reviewer 3**

In introduction or later in results part I would like to see comprehensive discussion on possible sampling artefacts. Statement on L112-113 that there is no possible break up just because modelling says so is not sufficient. For example, can the effect of scavenging of interstitial particles by ice crystals and droplets by estimated?

*This topic was the subject of Pekour and Cziczo (2011); we now include "A treatment of inadvertent transmission of particles smaller than the D50 as well as droplet and ice crystal breakup was considered by Pekour and Cziczo (2011); specifics for SPIDER are discussed in the following sections." in the introduction and an increased discussion in Section 2.*

How relevant are latex PSL spheres used for calibration with respect to different aerodynamic behaviour of ice crystals?

*Please see extensively expanded calibration tests with both droplets and ice.*

There are earlier studies showing (e.g. Fig 4 in Kupiszewski, 2016) that in various environments there are smaller ice crystals and INP particles than is the lower cut size of sizing OPC and cut off selection size of PCVI. Using instrument with lower cut off on both, residual and interstitial (PF) flow line is necessary to provide relevant quantitative characterization of the instrument.

*We have enhanced the measurements as much as possible with new instrumentation, including an APS. Regarding small particle transmission, please see previous comments that all PCVIs are subject to some level of inadvertent transmission, normally proportional to the number density of aerosol particles. Therefore, small particle transmission is expected, the goal is the reduce it as much as possible. Since the focus of these studies are larger particles the reduction of small particle artifacts was not a focus of this work.*

Also lower size cut off of the initial separation around 10 um does not cover full size of spectra of hydrometeors ( e.g. Patade, 2015)and this should be discussed in the manuscript how SPIDER can be possibly modified or combined with additional instrumentation to provide relevant information on how big fraction of population it actually sample.

*The field work from 2021-22 is now removed.*

References

Kupizsewski et al, JGR 2016, https://doi.org/10.1002/2016JD024894
Patade et al, JGR 2015, https://doi.org/10.1002/2015JD023375